# Fluorescence Guidance in Glioma Surgery: A Narrative Review of Current Evidence and the Drive Towards Objective Margin Differentiation

**DOI:** 10.3390/cancers17122019

**Published:** 2025-06-17

**Authors:** Matthew Elliot, Silvère Ségaud, Jose Pedro Lavrador, Francesco Vergani, Ranjeev Bhangoo, Keyoumars Ashkan, Yijing Xie, Graeme J. Stasiuk, Tom Vercauteren, Jonathan Shapey

**Affiliations:** 1Department of Surgical Interventional Engineering, School of Biomedical Engineering and Imaging Sciences, King’s College London, London SE1 7EH, UK; matthew.elliot@kcl.ac.uk (M.E.); silvere.segaud@kcl.ac.uk (S.S.); k.ashkan@nhs.net (K.A.); yijing.xie@kcl.ac.uk (Y.X.); tom.vercauteren@kcl.ac.uk (T.V.); 2Department of Neurosurgery, King’s College London Hospital NHS Foundation Trust, London SE5 9RS, UK; josepedro.lavrador@nhs.net (J.P.L.); francesco.vergani@nhs.net (F.V.); ranj.bhangoo@nhs.net (R.B.); 3Department of Imaging Chemistry and Biology, School of Biomedical Engineering and Imaging Sciences, King’s College London, London SE1 7EH, UK; graeme.stasiuk@kcl.ac.uk

**Keywords:** fluorescence, fluorescence-guided surgery, glioma, 5-ALA, PpIX, fluorescein, ICG, quantitative fluorescence

## Abstract

Brain tumours are difficult to remove. In part, this is because it is difficult for surgeons to accurately identify the boundaries between these tumours and the surrounding brain. Medications that cause tumours to fluoresce (emit light) during surgery were pioneered in brain tumours to help surgeons identify such tumour boundaries, and are now in regular use. We describe the evidence for the three medications currently used for fluorescence guided brain tumour surgery. We discuss how they work, how surgeons use them, the evidence for their accuracy and their effect on the outcomes of surgery. We highlight their limitations, and areas where further research could improve their use. We also explore how new work is aiming to expand the information that surgeons can collect from brain tumour fluorescence whilst also improving its accuracy. This includes the development of new systems to accurately measure the amount of emitted light (termed quantitative fluorescence), and the development of new fluorescent medications that are designed to overcome the limitations of current methods.

## 1. Introduction

Gliomas are heterogeneous, diffuse, infiltrative tumours that represent the most common and fatal group of primary brain tumours worldwide [1]. Since the introduction of the Stupp protocol of adjuvant radiotherapy and chemotherapy [2], gross total resection (GTR) has become the most significant modifiable determinant of prognosis for high-grade glioma (HGG) [3,4]. This must be balanced with the preservation of neurological function, emphasised by the increasing use of “deterioration-free survival” as a patient-centred outcome measure [5]. However, achieving a maximal, functional resection remains challenging, and despite modern surgical adjuncts, GTR is only achieved in between 15 and 67% of HGG resections [6].

The use of fluorescence for margin differentiation in glioma was first pioneered in 1948 [7] and, following a landmark Phase 3 trial in 2006 [8], has taken on an increasingly prominent role in neurosurgical oncology. Fluorescence-guided surgery (FGS) utilises the properties of molecules termed fluorophores to absorb electromagnetic radiation (excitation) and then emit light (emission) at a different wavelength. Ideal fluorophores must demonstrate a robust safety profile, have sufficiently long half-lives to allow intra-operative detection, and integrate into the surgical workflow. Crucially, they must selectively accumulate in pathological tissue and demonstrate sufficient diagnostic accuracy to guide resection. 5-Aminolevulinic acid (5-ALA) and fluorescein sodium (FS) are now routinely used for FGS in glioma. Indocyanine green (ICG), pioneered in vascular neurosurgery, has also demonstrated increasing utility with the development of novel “second-window” imaging techniques.

This review explores the evidence for these fluorophores as used in current practice, including the regulatory frameworks that govern their use. Emerging theories of immune-mediated action for common clinical fluorophores are discussed, along with the implications that fluorescence imaging may be able to provide real-time intraoperative insights into the tumour micro-environment, provided it can be imaged with sufficient spatial and spectral resolution. Whilst other recent reviews have focussed on technologies currently available in the clinic [9,10] or for specific subtypes of glioma [11], this review provides a more forward-looking perspective towards emerging FGS technologies, discussing these in the context of current surgical adjuncts, highlighting their potential, current limitations, and areas for further development.

FGS for glioma surgery is constrained by a reliance on subjective emission, i.e., the operating surgeon’s perception of fluorescence light. Commercially available equipment to mitigate this is detailed, including microscope-embedded excitation and emission filters. Other novel research techniques being developed for objective and quantitative fluorescence detection in glioma surgery are also explored, highlighting recent work to increase the sensitivity and specificity of FGS using quantitative fluorescence and derive novel information on tumour behaviour using its spectral signature. Finally, we review emerging development pipelines for novel fluorophores, including rationally designed molecules aimed at optimising both tumour targeting and signal-to-noise ratio. Significantly, we highlight the disappointing early results from in vivo trials of novel fluorophores within the context of the prolonged pathway to regulatory approval.

## 2. 5-Aminolevulinic Acid

5-ALA ($Gleolan^{TM}$, Medac, Wedel, Germany) is a non-fluorescent molecule involved in the heme biosynthesis pathway. It undergoes an intra-cellular, multi-step conversion to the fluorescent protoporphyrin IX (PpIX) before a rate-limited conversion to heme [12]. PpIX absorbs light in the blue range, with a maxima at 405 nm before emitting fluorescence with an emission maxima at 635 nm that can be visualised as red/pink fluorescence [13] (Figure 1).

### 2.1. Use Case

#### 2.1.1. Patient Population

Following a landmark Phase 3 trial [8], 5-ALA FGS was approved for HGG in Europe in 2007 and in 2017 in the USA [14]. Whilst widely used in HGG [15], off-label use in LGG has progressively increased, following emerging evidence that PpIX fluorescence in LGG can indicate areas of malignant transformation and act as a prognostic biomarker [16,17,18,19,20,21,22].

#### 2.1.2. Dose, Administration and Timing

Manufacturer guidance for 5-ALA recommends an oral dose of 20 mg/kg to be given 2–4 h prior to the induction of anaesthesia [23]. Dose guidelines were initially based on preclinical studies in rats [13,24] and human case series [25] before its use in the landmark Phase 3 trial [8]. A subsequent 2017 Phase 1/2 trial of twenty-one patients designed to test the efficacy of alternative dose regimes confirmed that maximal measurable fluorescence with minimal side effects was obtained using 20 mg/kg in HGG [26].

Limited evidence for optimal dosing has been published in LGG. Whilst the majority of trials use the established 20 mg/kg, a trial of 23 patients gave 10 LGG patients 40 mg/mL and showed a significant increase in PpIX concentration using this increased dose [27]. Whilst there was no increase in visible fluorescence, the need for further research into dose escalation in LGG is highlighted.

5-ALA is taken orally as a liquid suspension. Preclinical HGG studies in rats demonstrated peak fluorescence 6 h after oral administration [13]. Current recommendations are for administration 3–4 h prior to theatre to allow for transfer time, induction of anaesthesia, positioning, and surgical approach. It is recommended that an additional dose is given if surgery is delayed by 12 h or more. More recently, attempts have been made to systematically assess the temporal changes in PpIX fluorescence over time in humans, utilising pseudo-quantitative hyperspectral imaging [9,28,29]. In 68 HGG and 25 LGG patients, a later fluorescence intensity peak was observed between 7 and 8 h from 5-ALA administration, particularly at tumour margins where fluorescence can provide the most pertinent information for margin differentiation. Consequently, the authors recommended earlier dosing at 4–5 h prior to surgery.

#### 2.1.3. Side Effects and Safety

Reported side effects in 5-ALA trials have been limited to mild, transient liver enzyme elevation, photosensitivity, and nausea [8,30,31]. In the UK, a 24 h period of protection from sunlight is recommended given the risk of transient photosensitivity [15].

### 2.2. Equipment

5-ALA FGS requires an excitation light source incorporating the Soret band absorption of PpIX at 405 nm. A long-pass band filter at 440 nm is therefore used to filter reflected excitation light whilst allowing the passage of low-intensity 450 nm background light and PpIX emission at 635 nm. Commercial systems typically use proprietary algorithms to scale the reflected blue light to overcome background autofluorescence and non-specific “red” signal whilst highlighting PpIX emission [32]. As such, normal brain appears blue, whilst PpIX fluorescence appears as a solid red at the tumour core, fading to pink towards the margin [33] Figure 1. Several manufacturers, including Zeiss (Göttingen, Germany), Leica (Wetzlar, Germany), and Möller-Wedel (Wedel, Germany), provide comprehensive 5-ALA packages integrated into surgical microscopes [34]. More recently, groups have begun to develop surgical loupes optimised for 5-ALA fluorescence, along with advanced spectroscopic detection systems, discussed in more detail below **(5. Towards objective margin differentiation)**.

### 2.3. Mechanism of Action

5-ALA is metabolised to PpIX during the mitochondrial phase of the heme biosynthesis pathway before a final and rate-limited chelation with iron forms heme [12]. Two models have been proposed for PpIX accumulation in glioma.

#### 2.3.1. Increased Intracellular Synthesis and Retention

Classic models for 5-ALA induced tumour fluorescence are based on increased intra-tumoural synthesis and reduced efflux of PpIX [35] along with tumour-driven breakdown of the blood–brain barrier (BBB) [34,36], as illustrated in Figure 2.

Robust identification of the steps within the heme biosynthesis pathway responsible for tumour-specific PpIX accumulation has been challenging. Studies of protein expression in highly fluorescent tumour samples show an upregulation of CPOX and PPOX driving increased PpIX synthesis, which is then coupled with downregulation of the ABCG2 membrane transporter, reducing PpIX efflux [37,38]. Analysis of mRNA expression has shown inconsistent results, including an increase [39], no significant difference [38,39], and downregulation [38,40] of enzymes involved in PpIX synthesis. Reasons for the apparent decoupling of mRNA and protein expression remain unclear; however, the correlation of CPOX, PPOX, and ABCG2 expression with fluorescence provides a model for increased PpIX accumulation consistent with the well-characterised metabolic heme pathway.

#### 2.3.2. Immune Synthesis

An alternative model of PpIX accumulation highlights the role of the glioma immune micro-environment in driving PpIX fluorescence. Using paired Raman histology and two-photon excitation fluorescence microscopy in ex vivo samples from 115 patients, PpIX was demonstrated to accumulate preferentially in cells of myeloid origin, consistent with tumour-associated macrophages [41]. This builds on work using single-cell sequencing in glioma samples, showing significant PpIX fluorescence in infiltrating myeloid cells [42] and correlating with analysis showing stronger PpIX fluorescence in tumours with intense immune infiltration [43] and strong macrophage uptake of PpIX [44,45]. Such work points to a possible future role of PpIX in the investigation of the glioma immune micro-environment, linking to overall prognosis [46].

### 2.4. Evidence Base

#### 2.4.1. Diagnostic Accuracy

The core function of FGS is to facilitate reliable and accurate differentiation of tumours from surrounding parenchyma under operative conditions. This has been well studied for 5-ALA in HGG (Table A1). Studies utilising modern surgical microscope filters show sensitivity and specificity ranging from 84 to 98.5% and 29.4 to 100% [30,47,48,49,50,51,52], respectively. Given the documented challenges in calculating objective sensitivity and specificity from brain tumours, where truly random sampling is not possible [53], diagnostic accuracy is more precisely expressed as positive predictive value (PPV) and negative predictive value (NPV)—reported as 92–96% and 12.5–69%, respectively. It is important to note that 5-ALA fluorescence has been reported in non-pathological tissue, including abnormal brain [50] and inflammatory tissue [54]. Care must be taken to correlate fluorescence with anatomy, pre-operative imaging, and complementary surgical adjuncts such as neurophysiological monitoring.

As use of 5-ALA FGS increased in Europe, users began to explore the potential to display areas of intra-tumoural malignant transformation in patients with suspected LGG [55]. Case series exploring 5-ALA in radiologically suspected LGG have replicated the observation that focal areas of positive fluorescence, in an otherwise non-fluorescing tumour, are highly predictive of malignant transformation [17], with reported PPV ranging from 85% [16] to 100% [18]. These findings underpin the increasing use of 5-ALA off-label in radiologically suspected LGG, particularly when there is suspicion of malignant transformation.

Studies of 5-ALA in histologically confirmed LGG have shown more equivocal results. Rates of fluorescence for WHO Grade 2 glioma range from 0% in small cases series [16,56,57] to 25% and [58] 35% [59] in larger series. In the largest case series to date, including 82 WHO Grade 2 gliomas, 16% showed visible fluorescence using contemporary operating microscopes with fluorescence filter sets [19]. Interestingly, whilst these tumours were still classified as histologically WHO Grade 2, they showed significantly increased proliferation compared to non-fluorescent LGGs. A follow-up series of 79 patients showed visible fluorescence in 22% [22]. Such low rates of visible fluorescence have driven efforts to develop novel technologies for the visualisation of 5-ALA in LGG (see Section 5).

#### 2.4.2. Clinical Efficacy

Two Phase 3 randomised control trials have demonstrated that 5-ALA FGS improves extent of resection (EOR) in HGG (Table 1). The landmark 2006 RCT [8] showed an increase in HGG GTR from 36% using white light to 65% in 270 patients. This led to the regulatory approval and widespread uptake of 5-ALA in Europe. To address concerns about the lack of modern surgical adjuncts such as neuronavigation that were not available for use in the Stummer trial, a further RCT was conducted [31]. This, again, confirmed an increase in GTR for HGG, from 47.8% using white light to 79% using 5-ALA, in 147 patients. Other case series report significant variations in rates of GTR, ranging from 30 [60] to 100% [61]. These are explained to some extent by variations in the definition of GTR, the variable use of additional surgical adjuncts, tumour location (eloquent vs. non-eloquent), and the surgical learning curve. This presents an ongoing challenge to pooled analysis in FGS.

This heterogeneity, along with differences in adjuvant treatment regimens (particularly the uptake of the Stupp [2] protocol from 2005 onwards), rates of redo surgery, salvage chemotherapy, and patient selection, also leads to significant limitations in discerning current survival rates for patients with HGG. Whilst not powered as survival studies, neither Phase 3 RCT reached statistical significance for OS, whilst PFS at 6 months did significantly improve in the 2006 trial [8]. Changes to the histological diagnosis of HGG between 1998 and 2024, including four updates to the WHO CNS tumour classification system [62], also limit comparison between studies and may account for the wide range of reported survival. Notwithstanding these limitations, recent meta-analyses do seem to demonstrate that 5-ALA does improve both progression-free survival (PFS) and overall survival (OS). A recent meta-analysis of 1984 patients found that 5-ALA increased PFS vs. WL for 88.4% of patients and OS in 67.5% [63], adding to a previous meta-analysis suggesting a 1-month increase in PFS and a 3-month increase in OS [64].

**Table 1 cancers-17-02019-t001:** Randomised control trials assessing the effect of 5-ALA on clinical outcome.

Study	Study Type	n	EOR/GTR	OS	PFS	Complications	Notes
Picart 2023 [31]	Phase 3 RCT 5-ALA vs. WL	147 5-ALA: 67 WL: 69	GTR 5-ALA: 79% WL: 47.8%	24m 5-ALA: 30.1% WL: 37.7%	6m 5-ALA: 70.2% WL: 68.4%	Deficit at 3m 5-ALA: 13.2% WL: 12.9%	All SOC neuronav used. Post-op protocol RT + CTh (Stupp).
Stummer 2006 [8]	Phase 3 RCT 5-ALA vs. WL	270 5-ALA: 139 WL: 131	GTR 5-ALA: 65% WL: 36%	5-ALA: 15.2m WL: 13.5m	6m 5-ALA: 41% WL: 21% Median 5-ALA: 5.1m WL: 3.6m	NR	Neuro-navigation precluded. Post-op protocol recommended RT only. Industry sponsored.
Eljamel 2008 [65]	Prospective single-centre RCT	27 5-ALA: 13 WL: 14	GTR 5-ALA: 77% WL: 29%	NR	6m 5-ALA: 80% WL: 70% Mean 5-ALA: 52.8w WL: 24.2w	Neurology NR	Combination of FGS and Photofin (R) used in 5-ALA group. Post-op protocol for RT alone. n = 7 also received CTh.

Abbreviations: EOR: extent of resection; GTR: gross total resection; OS: overall survival; PFS: progression-free survival; WL: white light; SOC: standard of care; RT: radiotherapy; CTh: chemotherapy.

Despite limited visible fluorescence, 5-ALA may also act as a prognostic biomarker in LGG. An analysis of survival in LGG found shorter PFS in tumours demonstrating intra-operative fluorescence [22,66], correlating with the evidence suggesting visible fluorescence correlates with increased proliferation [19].

### 2.5. Limitations

Use of 5-ALA for margin detection is limited by imaging geometry, high cost, reliance on subjective interpretation, and low emission in LGG. Efforts to address these limitations through the development of quantitative fluorescence detection systems are detailed in **(5. Towards objective margin differentiation)**.

The infiltrative nature of glioma means low levels of fluorescence are typically seen at the tumour margin [67] and biopsies from the non-fluorescent boundaries often contain neoplastic cells [51]. Recurrence typically occurs at resection cavity margins, even with complete resection of visible fluorescence [68,69]. A reliance on subjective interpretation of low levels of visible fluorescence at tumour boundaries contributes to the low NPV of 5-ALA fluorescence in HGG, introduces inter-observer variability to 5-ALA FGS [70], and limits the use of 5-ALA in LGG. Despite fluorescence highlighting tumour beyond the margins of MRI enhancement [71], correlation of visible 5ALA fluorescence with peri-tumoural fluid attenuated inversion recovery (FLAIR) signal change in HGG is currently unclear [72] As evidence emerges on the use of supra-marginal resection in glioma [73], an ongoing prospective evaluation of the value added by FGS is essential [74]. Surgeons must also be alert to the small but clinically significant rates of false-positive 5-ALA fluorescence, typically found in areas of high macrophage infiltration [75,76,77,78]. These issues have underpinned a drive in recent years to develop systems able to quantify fluorescence emission in FGS.

5-ALA FGS also imposes a significant cost burden on neurosurgical units. Proprietary microscope FGS systems typically incur significant upfront costs, and use of 5-ALA has been estimated to cost EUR 9021 per quality-adjusted life year (QALY) in Spain [79] and up to USD 12,817 per QALY in the USA [80]. In the UK, the cost of using 5-ALA is estimated at up to GBP 1,000,000 per neurosurgical unit per year [15].

### 2.6. Regulatory Issues

5-ALA’s use as an optical imaging agent in HGG was approved by the European Medicines Agency in 2007 and a decade later by the FDA in 2017. It remains the only fluorescent agent licenced for use in glioma surgery. Off-label use in LGG is common throughout Europe, Japan, Korea, and North America [14]. Delayed approval for 5-ALA use in the USA was a result of the FDA classifying 5-ALA as a cross-over therapeutic and imaging agent. This requires evidence that the agent not only results in increased rates of GTR but also confers increased survival benefits [81]. However, both FDA and EMA scientific reviews note a lack of definitive evidence for the selection of dose, timing of administration, and definition of “border” in HGG [82,83].

### 2.7. Emerging Use Cases

5-ALA was pioneered for tumour differentiation, but research groups have since explored its photo-sensitising potential, complex spectral behaviour, and localisation of PpIX in the glioma micro-environment. Potential novel use cases have emerged including use in photodynamic therapy, real-time intra-operative tumour classification, and prognostication.

PpIX exposed to adequate light reacts to produce cytotoxic single oxygen radicals. Case reports using PpIX photodynamic therapy for the treatment of small glioma lesions have shown early promise [84,85]. A Phase 1 clinical trial of 10 patients was commenced in 2017 and has completed recruitment—Intraoperarive Photodynamic Therapy for Gliobastomas (INDYGO) [86]. A recent 5-year report noted a median follow-up of 23 months, with Kaplan–Meir estimated median overall survival of 23.4 months, a 12-month OS of 80%, and 12-month PFS of 60% [87].

The potential for 5-ALA to provide insight into glioma pathology and the molecular profile of tumours was noted during early investigations in LGG [16]. As a deeper understanding of the spectral complexity of PpIX fluorescence in glioma has developed [88,89], along with technologies for intra-operative spectral characterisation of PpIX [90], the possibility of extracting neuropathological information from the PpIX emission spectra has emerged. Distinct, environmentally driven photochemical states of PpIX emitting at 620 nm and 635 nm have been unmixed from glioma fluorescence, and the ratio is hypothesised to correlate with markers of malignancy [89,91,92]. However, despite this promising evidence, it remains unclear to what extent the 620 nm peak represents PpIX versus other 5-ALA metabolism intermediaries [93]. Improved, spectrally unmixed datasets paired with gold-standard molecular histology from human glioma are required to understand the information on tumour biology contained in 5-ALA-derived fluorescence emission. This work has the potential to dovetail with emerging evidence of mechanisms of PpIX accumulation to provide an insight into the tumour immune micro-environment and molecular pathology [41]. As evidence emerges for the use of personalised oncological treatment in HGG [94], this holds the potential to provide critical insights into the behaviour of heterogenous tumour tissue left at function resection boundaries [95].

## 3. Fluorescein

Fluorescein is a fluorescent small molecule available in 1 g vials costing approximately 5 Euros (EUR 5.70, GBP 4.30) per vial [96]. Following intravenous administration, it weakly binds to plasma proteins such as albumin and accumulates extracellularly in areas of BBB breakdown. Fluorescein absorbs light in the range of 465–490 nm with a maxima at 480 nm, before emitting a yellow/green fluorescence at 520–690 nm with a maxima at 525 nm [97,98,99,100] as illustrated in Figure 3.

### 3.1. Use Case

#### 3.1.1. Patient Population

In 1948, Moore described the utility of fluorescein for the differentiation of “intra-cranial malignancies”—ranging from meningioma to glioblastoma [7]. In glioma surgery, use of fluorescein has typically been confined to HGG, where breakdown of the BBB is expected [101]. A 2021 meta-analysis of 336 HGG patients [102] concluded that fluorescein-guided surgery led to comparable rates of GTR vs. 5-ALA, with a significant increase vs. white light alone. Whilst there has been an increasing appetite to explore the role of FGS in LGG, a 2023 systematic review found limited evidence of the use of fluorescein in LGG [21]. Indeed, only 13 WHO Grade 2 patients had undergone surgery using fluorescein, of which 38% showed visible fluorescence [21].

#### 3.1.2. Dose, Administration and Timing

Fluorescein dose for glioma surgery remains unstandardised [103]. Whilst Moore used a fixed, high dose of 1 g, studies published prior to the advent of selective fluorescence microscopy typically report effective doses of 15–20 mg/kg [102,104,105]. The development of improved visualization techniques including the microscope filters described below have led to a progressive reduction in the apparent effective dose, with the largest phase II study to date utilising 5–10 mg/kg [106]. A 2024 prospective study of 60 HGG patients included 30 treated with 1 mg/kg—and concluded such “ultra-low” doses may provide comparable tumour discrimination to 5 mg/kg with reduced side effect profile [107].

Fluorescein is water soluble and given intravenously, but there is a lack of data on the optimal timing administration for use in glioma surgery [9]. Most authors report administration at the time of anaesthetic induction, approximately 60–90 min prior to tumour resection, to allow the washout of the unbound form and clearance from non-tumour tissue [106,107,108]. Investigation of fluorescein pharmacokinetics [109] has led to calls for administration to be pushed back further—at 2–4 h prior to tumour resection, to ensure washout of non-specific fluorescence resulting from unbound fluorescein [9,101]. Other authors, however, advocate administration around the time of dural opening (i.e. within 10 min of commencing tumour resection) [104,110]. Whilst this mirrors the use of fluorescein in retinal angiography, it has been criticised because unbound fluorescein may cross the BBB in the acute administration phase. There is also a risk of “contrast leak” from vessels damaged during surgery [111]. Both of these phenomena introduce the possibility of non-specific fluorescence and reduced signal-to-noise ratio [112].

#### 3.1.3. Side Effects and Safety

Fluorescein has been widely studied in ophthalmology and is considered to have an excellent safety profile. Adverse events are reported in 3.3% of patients [113,114]. Allergic reactions can include respiratory, cardiac, anaphylaxis, and allergic skin testing before surgery is recommended [115]. Care must be taken to avoid extravasation during injection. Yellow discolouration of the skin and sclera is common and typically resolves in 24 h. Contraindications are the use of beta blockers along with renal or hepatic insufficiency.

### 3.2. Equipment

When used at 20 mg/kg, fluorescein in HGG is visible to the naked eye when excited with blue light [102]. A number of proprietary microscope filters have been developed for use with fluorescein, all of which are tuned to provide excitation light in the 460–500 nm range and for observation in the 540–590 nm range. The improved visualisation provided by these filters has driven work to reduce the dose administered as described above and contributed significantly to increased use of fluorescein in HGG [116]. These systems also allow “mixing” of red and blue light using proprietary algorithms to improve visualisation of non-fluorescent tissue, allowing non-pathological structures to be identified and preserved whilst operating in fluorescence mode [117]. Given the high cost of these systems, early work has begun in developing headlight and loupe fluorescein detection kits for use in environments where resources are constrained [118].

### 3.3. Mechanism of Action

Upon administration, fluorescein undergoes dose-dependent binding to serum proteins and distributes through the circulation and interstitial spaces [119]. Unbound fluorescein crosses the BBB in mouse models, and high concentrations are found throughout the brain following human equivalent dosing [109]. Non-specific fluorescence from this unbound component peaks 15–30 min following administration and progressively decreases thereafter. The bound fraction shows minimal transport across the intact BBB but crosses readily into the brain at areas of BBB disruption and thereby accumulates in the extracellular space [106,120], accentuated by the enhanced permeability and retention effect in tumour tissue [101]. Areas lacking an intact BBB, such as dura and choroid, are intensely fluorescent.

### 3.4. Evidence Base

#### 3.4.1. Diagnostic Accuracy

Fluorescein’s mechanism of action and variability in dose, equipment, and timing has led to concerns about its diagnostic accuracy that have hindered uptake in some centres [121,122]. Using current systems, biopsy-based sensitivities in small, non-randomised prospective studies range from 75.6 to 93.5%, with specificity ranging from 75 to 90.9% (Table A2). When biopsies are limited to contrast-enhancing tumour, the lower range increases to 82.8% and 82.6% for sensitivity and specificity, respectively [110]. Sensitivity and specificity do not appear to correlate with dose, and indeed lower doses with adequate filtering may reduce non-specific fluorescence that can lower diagnostic accuracy [107]. Relatively few studies have calculated PPV and NPV for fluorescein in glioma surgery. FLUOGLIO utilised a subset of 50 biopsies in 13 patients to calculate a PPV of 80.8 and NPV of 79.1. Other groups calculated PPV and NPV using inspection of the surgical cavity compared with post-op MRI imaging in 43 HGG patients. These results suggesting a PPV of 100 and NPV of 81 are encouraging; however, they are limited by constraints of post-operative MRI in the detection of tumours [123].

Whilst published values for the diagnostic accuracy of using fluorescein in glioma surgery are comparable with 5-ALA, the quality and volume of evidence underpinning these calculations are more circumspect. This is particularly significant given the low number of studies that report the clinically relevant parameters of PPV and NPV. Diagnostic accuracy data in LGG is limited and appears significantly lower than for HGG. Ling et al. [107] reported that 42 biopsies taken from 7 patients undergoing fluorescein-guided LGG surgery showed a sensitivity of 64.3–66.47 and specificity 57.1–61.1% [107]. Although potentially useful for highlighting areas of contrast enhancement, the evidence in LGG currently does not support its use outside of a research setting.

#### 3.4.2. Clinical Efficacy

Fluorescein is well established as safe in glioma surgery; however, to date, no Phase 3 RCTs have been conducted to evaluate its impact on rates of glioma resection and survival. Despite this, logistical and cost advantages of fluorescein have led to a sustained attempt to establish its efficacy. A meta-analysis [102] pooled data from 331 patients across 21 studies to demonstrate a GTR of 81%, comparable with that seen in the 5-ALA Phase 3 trials. The authors note, however, that 11 studies do not include a control group, and there is significant heterogeneity in the definitions of GTR, timing, and dose of fluorescein and in the grading of HGG. Furthermore, a significant number are non-randomised, retrospective studies. Only 7 of the 21 included data on overall survival, and 2 included data on PFS. None reported mean overall survival time.

Five prospective studies reporting survival data show GTR ranging from 74 to 90% (Table 2). Median PFS in three studies ranges from 7 to 12 months, with overall survival from 12 to 16 months. Other retrospective datasets have recorded GTR rates of 81.1–97.4% [105,124,125,126]. Median PFS was 9.2 months in the largest reporting dataset to date [125].

It is important to note the significant limitations to this evidence base. Whilst in recent years protocols have become more standardized, there remains variability in the dose and timing of administration that hinders data pooling. A large proportion of the literature does not include case-matched controls or relies on historic control data with the inherent risk of bias. As with all surgical studies in glioma, comparison of survival over time is hindered by changes to the postoperative treatment available to patients, and a significant number of published fluorescein studies do not report survival.

### 3.5. Limitations

Fluorescein is limited by subjective interpretation, a low signal-to-noise ratio at tumour margins—worsened by the potential for surrounding oedema [122]—and low emission in tumours such as LGG where BBB breakdown is limited [21]. An inherent limitation of fluorescein lies in its non-specific mechanism of action, showing extracellular accumulation in areas of BBB breakdown rather than being specifically targeted to tumour tissue [127].

Care must also be taken with timing the administration of fluorescein given the propensity of the unbound component to freely cross the BBB in the early period following injection.

### 3.6. Regulatory Issues

Fluorescein is approved by both the FDA and the National Institute for Health and Care Excellence (NICE) for ocular angiography. The Italian drug agency has approved fluorescein as a fluorescence tracer for aggressive CNS tumours [128]; however, this is not the case elsewhere in Europe or the USA. All use in low-grade gliomas remains off-label.

**Table 2 cancers-17-02019-t002:** Prospective trials assessing the effect of FS on clinical outcome.

Study	Study Type	n	EOR/GTR	OS	PFS	Complications	Notes
Ling 2024 [107]	Prospective non-randomised	90 FS LD: 30 FS StD: 30 Cont: 30	GTR% FS LD: 90 FS StD: 86.7 Cont: 66.3	—	6m% FS LD: 90 FS StD: 86.7 Cont: 66.3	Dependent 6m: FS LD: 10% FS StD: 13.3% Cont: 36.7%	LD: 1 mg/kg StdD: 5 mg/kg Adm: Post-intubation Historic single-centre control
Falco 2019/2023 [128,129]	Prospective non-randomised	279 HGG: 128 GBM: 93 LGG: 11	GTR (%) HGG: 74.2 GBM: 82.8	Median (m) GBM: 16	Median (m) GBM: 12	No adverse reactions	5 mg/kg Adm: After induction No LGG fluorescence Retrospective survival analysis
Acerbi 2018 [106]	Prospective, multicentric phase II, FLUOGLIO	46	GTR FS: 82.6 5-ALA: 32 WL: 36	Median (m) 12	Median (m) 7 6m: 56.6 12m: 15.2	No FS related AE KPS returned to baseline 3m	5–10 mg/kg Adm: After induction 30 adjacent to eloquent areas STUPP [2] completed in 20%
Chen 2012 [130]	Prospective non-randomised	22 FS: 10 Control: 12	GTR (%) FS: 80 Control: 33.3	—	Median (m) FS: 7.2 Control: 4.8	No significant difference in KPS between groups	15–20 mg/kg WL guided Adm: Following dural opening HGG: 11 LGG: 11
Koc 2008 [131]	Prospective non-randomised	80 FS: 47 Control: 33	GTR (%) FS: 83 WL: 55	Median (w) FS: 44 WL: 42	-	No significant difference in KPS between groups	20 mg/kg WL guided Adm: Prior to dural opening

Abbreviations: EOR: extent of resection; GTR: gross total resection; OS: overall survival; PFS: progression-free survival; M: months; W: weeks; LD: low dose; StdD: standard dose; Cont: control; Adm: administration; KPS: Karnofsky score; RCT: randomized controlled trial; WL: White light.

## 4. ICG

Indocyanine green (ICG) is an amphiphilic tricarbocyanine molecule that absorbs light at 778–806 nm before emitting in the near-infrared (NIR) window at 835 nm, with a secondary peak in the second NIR (NIRII) or shortwave infrared (SWIR) window from 1000 to 1700 nm [132]. This allows improved tissue penetration, simultaneous white light, and fluorescence imaging and significant improvements in signal-to-noise ratio. In glioma surgery, ICG has been used for angiography, as discussed in Section 4.1, and tumour differentiation using the “second window” effect, as discussed in Section 4.2.

### 4.1. ICG Angiography

ICG is established in intracerebral angiography. A 2003 landmark study demonstrated a tight correlation between intraoperative fluorescence and postoperative angiography in aneurysm and dural arteriovenous fistulas, and this technique is now routinely used in vascular neurosurgery [133]. Following this success, groups have exploited differences in the vascular structure of gliomas and the surrounding parenchyma with the aim of reducing postoperative complications resulting from vascular injury and improving tumour differentiation Figure 4.

#### 4.1.1. Dose, Administration and Timing

When used for angiography, ICG is administered as a standard intravenous bolus dose of 0.2–0.5 mg/kg, immediately prior to imaging in the NIR window [134].

#### 4.1.2. Mechanism of Action

ICG rapidly binds to plasma proteins and is confined to the vascular compartment. This allows high-fidelity fluorescence imaging of the intracranial vessels during surgery. This “first-window” fluorescence is rapidly cleared by the liver and excreted in bile—with a half-life of 10–15 min, allowing multiple imaging runs during a single procedure.

#### 4.1.3. Evidence Base

ICG angiography was first performed for glioma in 1996 [135]. Doses of 2 mg/kg were used in 9 glioma patients to demonstrate a dynamic difference in perfusion between tumour tissue and parenchyma. In 2011, two groups used the technique to demonstrate tumour-specific vascular patterns, identify adjacent vessels for preservation, and highlight patients with a high post-operative risk of ischaemia [136,137]). A 2018 retrospective review of 10 gliomas allowed the identification of arterialised and thrombosed veins in 3 cases, allowing disconnection of the main venous collector to restore normal venous flow [138].

A 2015 study expanded the technique from the protection of vessels to tumour differentiation [139]. ICG was used with 5-ALA to demonstrate differing vascular architecture in three distinct “zones”, comprising tumour core, transition zone, and surrounding parenchyma. As noted by the authors, this indirect technique introduces the significant risk of resection of functional tissue and has not been evaluated using standard measures of diagnostic accuracy or validated in larger patient cohorts.

### 4.2. ICG Second-Window Tumour Differentiation

Prolonged ICG fluorescence and reduced tumour clearance [135] was used for the first time in humans for tissue differentiation based on the “second window effect” in 2016 [140]. This is a distinct traditional first-window use of ICG, where fluorescence imaging is performed immediately following injection when the fluorophore is confined to the vascular compartment. Instead, ICG is administered at higher doses up to 24–48 h prior to imaging, which is performed when the fluorophore has circulated throughout the body and accumulated in pathological tissue, as discussed below.

#### 4.2.1. Mechanism of Action

Second-window ICG is underpinned by the enhanced permeability and retention (EPR) hypothesis. The combined increase in vascular permeability and impairment of lymphatic drainage allows the accumulation of ICG in tumours such that it can be visualised 24–48 h after administration [141].

#### 4.2.2. Dose, Administration, and Timing

Second-window ICG requires doses higher than the FDA maximum recommended human dose [132]. Initial recommendations of 5 mg/kg [140] were based on pre-clinical studies using murine models, backed by first-in-human trials for thoracic carcinoma showing at doses below a 5 mg/kg tumour-to-background fluorescence ratio, and perceived fluorescence was low [142]. Doses up to 10 mg/kg did not improve subjective fluorescence perception. A subsequent dose-escalation study in 45 patients with thoracic malignancy using doses of 1–5 mg/kg and 2–3 mg/kg was found to give acceptable signal across all subtypes tested [143]. This led to a dose decrease, initially to 2.5 mg/kg [144] and further to 1 mg/kg as detection camera systems have improved [145,146]. To date, a dose-escalation/optimization study has not been performed for human glioma surgery using second-window ICG.

Preclinical studies demonstrate a peak tumour-to-background ratio at 24 h from administration [142], with mouse GBM models showing a broad plateau up to 48 h [147]. Current evidence indicates second window fluorescence is not significantly time dependent from 6 to 48 h; however this is yet to be robustly tested in humans.

#### 4.2.3. Evidence Base

No large studies have investigated the use of second-window ICG on survival in glioma. A small study of 15 GBM patients undergoing FGS vs. 18 controls suggested an improved PFS of 9 months vs. 7 months (*p* = 0.0002) [146]. Studies of diagnostic accuracy have typically been small, have not included control arms, and are limited by the constraints of tumour sampling in vivo. These indicate a high sensitivity with low specificity. This is improving as more sophisticated detection systems, development of AI post-processing algorithms, and lower dose protocols are developed (Table A3). Published sensitivity ranges from 85.7 to 100% [140,145,146,147,148]. Specificity ranges from 25 to 56% in early studies using 5 mg/kg [140,147,148], up to 91.36% in more recent, small studies [146]. Due to this limited evidence base, the use of second-window ICG in tumour surgery remains confined to the research setting.

### 4.3. Side Effects and Safety

Side effects from ICG are rare, and its use is well established in hepatic, ophthalmic, and neurovascular imaging. Nausea, vomiting, and skin rash are reported in 0.2% of cases, with hypotension, arrhythmia, and anaphylactic shock in 0.05% [149].

### 4.4. Equipment

ICG fluorescence is emitted in the infrared spectrum and is not visible to the naked eye. Early studies used stand-alone, exoscope-based IR imaging systems that were validated for use in the operating theatre (Visionsense Iridium, Visionsense, Philadelphia, Pennsylvania) [140,147,148]. More recently, commercial IR imaging systems integrated into neurosurgical microscopes have become well established for imaging ICG fluorescence in neurovascular surgery, including Infrared800 and Flow800 (Zeiss, Göttingen, Germany) and Glow800 (Leica, Wetzlar, Germany), all of which can be adapted to utilise second-window imaging of tumours. More recent work evaluating imaging in both the SWIR and NIR II windows utilise either multispectral imaging systems [146] or specialised, filtered CCD cameras with laser excitation [145], which remain limited to the research setting.

### 4.5. Limitations

ICG in glioma remains a research tool. Promising work identifying tumour vessels has not been expanded to show a robust impact on either resection, complication rates, or survival. As novel outcome measures such as deterioration-free survival take increasing prominence in the neurosurgical literature [5], this type of technique aimed at reducing complications warrants further study.

Second-window ICG imaging is limited by its non-specific mechanism of action and low specificity. Progress has been made in this regard through the use of improved imaging modalities, including NIR II fluorescence and the use of convoluted neural networks for imaging analysis [145,146]. Work evaluating these systems in larger, randomised patient groups has the potential to significantly expand the neurosurgical armamentarium.

### 4.6. Regulatory Issues

FDA approval was granted in 1956 for the use of ICG in cardio-circulatory and hepatic investigation. This was expanded to ophthalmic angiography in 1975. The Leica FL800 system received FDA approval for ICG-guided cerebral angiography in 2006, and the first Zeiss system was approved in 2008 (K100468) [150]. It is important to note that use of cerebral angiography for tumour differentiation remains off-label, and FDA approvals remains device specific. Second-window ICG imaging remains off-label and limited to a research context.

## 5. Towards Objective Intra-Operative Fluorescence

FGS systems in clinical use provide a qualitative fluorescence image using the operative microscope, which is interpreted by the operating surgeon. The emitted light is passed through wavelength-specific filters to the surgical optics, and surgeons use this visible light to make judgments on the tissue under interrogation. Reliance on visible fluorescence generates positive predictive values of >90%, however is limited by low emission at infiltrative margins, and in LGG. As such, negative predictive values are as low as 35% [9]. Attempts to improve sensitivity through increasing fluorophore dose have been trailed using 5-ALA in LGG, with rates of visible fluorescence remaining limited to 60% [27].

A number of novel imaging systems have been released with the aim of improving the sensitivity and specificity of FGS by improving imaging geometry. Systems such as the Nico Myriad Spectra (Nico Corporation, Indianapolis, IN, USA) allow the introduction of excitation light immediately adjacent to the tissue under interrogation, with early experience within our group suggesting this approach can be particularly useful during minimally invasive approaches [151,152]. This remains to be validated in a prospective, multicentre trial. Similarly, a small retrospective study of 10 patients with HGG has suggested use of an exoscope may allow for improved fluorescence visualization versus an operative microscope [153]. A number of groups have developed surgical loupes modified for FGS with promising initial results, suggesting visible fluorescence intensities up to 9.9 times higher than when using the operative microscope [118,154,155].

These techniques remain constrained by subjective analysis of visible fluorescence. This limitation underpins the ongoing drive to develop objective, sensitive and specific imaging systems for the evaluation of glioma infiltration utilising established fluorophores.

### 5.1. Steady-State Fluorescence Spectroscopy

Fluorescence spectroscopy involves the collection of spectrally (i.e., wavelength-specific) resolved information either intra-operatively or ex vivo. This tissue-specific “optical fingerprint” can be used for tissue differentiation and non-invasive interrogation of tumour metabolism. This approach can either be based on the measurement of emitted photons, so-called “intensity” and “steady-state” spectroscopy, or the measurement of fluorescence lifetime “time-resolved fluorescence spectroscopy”. Time-resolved fluorescence spectroscopy measures the time between absorption of an excitation photon and emission of a fluorescence photon and is discussed in Section 5.3.

Early steady-state spectroscopy studies used point probes encompassing laser excitation with spatially paired detectors without exogenous fluorescence labels. These demonstrated that when excited at 360 nm, the emission from endogenous fluorophores such as NADH, FAD, and Lipofuscin was significantly different in glioma compared to non-infiltrated brain tissue samples, both ex vivo and in vivo [156]. This technique, described as label-free spectroscopy, has been trialled in pilot studies involving 39 patients, with reported sensitivities of 100% and specificities of 76%. Subsequent works report sensitivities and specificities of up to 94% and 93%, respectively [157]. These techniques also allow the identification of endogenous fluorophores involved in metabolism, with the potential of generating real-time insights into tumour metabolic pathways [158].

Label-based spectroscopic fluorescence for glioma surgery is underpinned by the hypothesis that fluorophores accumulate in LGG and the IM at levels too low for visual detection but in quantities significantly higher than non-neoplastic tissue. Point probe systems detecting “raw” fluorescence emission of PpIX has demonstrated a PPV of 96.2%, with an NPV of 39.5% [47].

### 5.2. Quantitative Fluorescence

Whilst an improvement on basic visual interpretation, spectroscopy utilising raw fluorescence emission remains qualitative in nature. Emission in complex biological systems such as glioma tissue is affected by a combination of optically distorting effects, the presence of overlapping endogenous fluorophores, and the photochemical state and quantum efficiency of the measured fluorophore. Objective, quantitative fluorescence (QF) spectroscopy requires the correction of each of these factors. The majority of QF work has been performed using 5-ALA, and a brief summary of these techniques is given below. The interested reader is referred to Valdes et al.’s [159] dedicated review article for further reading [159] on this topic.

#### 5.2.1. Optical Distortion Correction

Fluorescence excitation and emission in tissue undergo a non-linear distortion as light is absorbed and scattered [159]. These properties are highly variable in glioma [160] and must be corrected to allow quantitative fluorescence. One method for this involves probe-based spectroscopy capable of measuring diffuse reflectance, coupled with computational models of light propagation alongside fluorescence emission [161,162,163,164,165]. Wide-field methods utilise highly selective cameras with tuneable spectral filters or hyperspectral cameras. These allow for spectral scanning and reconstruction of spatially and spectrally resolved data with optical correction based on diffuse reflectance [159,166,167]. Both are limited by a lack of validation in tightly paired optical and fluorescence readings and a reliance on diffuse reflectance for correction. Probe measurements are further constrained by a narrow detection area. Conversely, wide-field scanning techniques are affected by delays in image acquisition. More recent hyperspectral “snapshot” and “light-field” systems can overcome the issue of imaging delay by capturing spectral and spatial information simultaneously in a mosaic pattern at the cost of spatial and spectral resolution [168].

#### 5.2.2. Fluorescence Unmixing

Following optical distortion correction, the calculated spectra comprise “tumour” fluorophore emission coupled with overlapping endogenous fluorophores such as free NADH, oxidised flavins, and lipofuscin. Attempts to isolate tumour-specific fluorescence have been performed with 5-ALA, using published datasets for PpIX spectra, whilst empirically calculating the spectra of auto-fluorescence constituents using known emission peaks [91]. The pre-determined constituents are then fitted to the measured spectra to extract PpIX-specific fluorescence. This represents a significant step forward; however, the method remains limited by the semi-empirical nature of auto-fluorescence characterisation and the need for high spectral resolution to be able to implement this technique in vivo.

#### 5.2.3. Emission Form

QF requires an understanding of fluorophore behaviour in the glioma micro-environment. PpIX exhibits two emission forms in vitro, characterised by primary emission peaks at 620 nm (PpIX_620_) and 635 nm (PpIX_635_). These forms differ in quantum efficiency and are thought to represent different aggregations [169]. The presence of both forms has been demonstrated in glioma tissue, with the ratio (PpIX_635/620_) hypothesised to provide an insight into malignancy [88,89]. It is uncertain whether other tetrapyrrole intermediaries within the heme biosynthesis pathway, such as coporphyrinogen III or uroporphyrinogen III, contribute to the 620 nm peak observed in vivo [93], further hindering accurate quantification. To date, hardware constraints have hindered the collection of data with sufficient spatial and spectral resolution for unmixing in vivo without introducing unacceptable delays in the surgical workflow; however, this remains a promising area for development.

### 5.3. Time-Resolved Fluorescence Decay

An alternative to spectral fluorophore characterisation is the use of fluorescence lifetime. This is a photophysical parameter that represents the energy relaxation dynamics of individual fluorophores. On a molecular level, fluorescence lifetime represents the average time a molecule spends in an excited state after absorption of excitation light before the emission of fluorescence. These measurements are largely independent of excitation intensity and allow unmixing of fluorophores in the temporal dimension. Lifetime has been explored in neuro-oncology to investigate tumour metabolic status [170], and more recently PpIX content [171,172,173] and differentiation of PpIX [174] from endogenous fluorophores [175]. Acquiring lifetime measurements is challenging to translate into the operating theatre, requiring a high-frequency pulsed laser for fluorophore excitation, a high precision, gated detector with high sensitivity to interference, and an extended acquisition time. Despite its limitations, attempts have been made to engineer systems that integrate into the surgical workflow [173] by pairing readings with raw fluorescence emission, albeit limited by a lack of optical distortion correction and limited integration.

### 5.4. Confocal Laser Endomicroscopy

Confocal laser endomicroscopy (CLE) uses sodium fluorescein as a contrast agent to generate intraoperative histological images in near real time—thus avoiding the challenges relating to non-specific targeting and complex fluorophore/tissue interactions as detailed above. Commercial CLE systems such as the Zeiss CONVIVO have recently been approved by the FDA.

To date CLE in glioma has only been used as a diagnostic tool [176]. The first prospective, multi-centre phase II non-inferiority study investigated the diagnostic accuracy of CLE vs. frozen section (FS) in 210 patients—86 of which were glioma [177]. This reported a CLE accuracy of 87% vs. 91% FS, below the threshold for non-inferiority (*p* = 0.367)—however a significant learning curve for CLE and higher diagnostic accuracy for tumour specimens was apparent. Ongoing improvement in diagnostic accuracy with wider uptake of the technique remains possible. However, the processing time of CLE was significantly faster than FS—with a median processing time of 3 min and 27 min, respectively. A similar study remains recruiting at the time of writing (Bern, Switzerland, CLEBT, NCT04280952 accessed 16 January 2025). Whilst a use of CLE in guiding marginal tumour resection has been hypothesised, to date there are no studies the authors are aware of assessing the impact of CLE on extent of resection or survival in glioma and this exciting use case remains to be fully explored.

### 5.5. Future Prospects

As discussed above, the development of novel imaging technologies has made objective fluorescence mapping in glioma FGS a viable possibility over the last 10–15 years. At present, these techniques are confined by the inherent trade-off between spatial resolution, spectral resolution, and acquisition time [178,179]. As technologies such as hyperspectral imaging rapidly improve, these hardware constraints are being progressively overcome. This is eloquently highlighted by the astounding progression from large, bulky spectral scanning systems with acquisition times in minutes [180] to stand-alone and microscope-integrated systems that acquire hypercube images in less than a second [168]. This hardware development, combined with the novel signal processing techniques described above holds the potential not only to improve the positive and negative predictive value of FGS at the tumour margins, but to unlock critical prognostic data present in the detailed spectral signature of current fluorophores to inform surgical decision making in real time.

## 6. Novel Fluorophore Development

Current fluorophores are limited by non-specific mechanisms of action, overlapping endogenous autofluorescence, depth of tissue penetration, emission/excitation wavelengths that preclude simultaneous white-light and fluorescence imaging, and distortion by commonly present compounds such as haemoglobin. To overcome these limitations, a number of novel fluorescence agents are currently in development. These have the dual aim of improving the intensity and visibility of tumour-associated fluorescence and improving specificity through improved tumour targeting. In 2024 approximately 40 fluorescence agents were under investigation in 85 clinical trials across all tumours in the United States alone [181]. The development of novel fluorophores focuses on two key aspects of FGS: tumour targeting to improve specificity and emission detection to improve sensitivity.

### 6.1. Tumour Targeting

Improving fluorescence targeting can utilise both “activatable” fluorophores, where an aspect of the tumour environment acts either to remove a fluorescence quenching moiety [182] or induce an activating structural change [183] alongside the targeting of receptors overexpressed in glioma. This has been attempted in Phase 1 clinical studies using humanised, monoclonal antibodies [184]; smaller antibody fragments aimed at improving penetration of the BBB [185,186]; and rationally designed small molecules [187] conjugated to existing fluorophores (Figure 5).

Use of a modified chlorotoxin peptide conjugated to ICG for tumour targeting (Tozuleristide, BLZ-100, Blaze Bioscience, Seattle, WA, USA) showed promise in a Phase 1 study [189], with optimal window dose showing a sensitivity of 82% and specificity of 89%. This remains the only novel fluorophore to complete a Phase 2/3 clinical in glioma, with equivocal early results posted in December 2024. In 105 subjects, sensitivity was 79.8%; however, specificity for equivocal tissue was 34.7% (NCT03579602 accessed 16 January 2025). A full breakdown of the study is yet to be published in the peer-reviewed literature.

Two Phase 2/3 trials investigating Panitumumab for EGFR targeting are ongoing (NCT04085887, not yet recruiting, accessed 16 January 2025, NCT03510208 estimated completion 25 December, accessed 16 January 2025). Seven Phase 1 trials have been completed investigating targeted fluorophores, including BLZ-100 [189,190], the GRPR-targeted $\hat{6}$8Ga-IRDye800CW-BBN [191,192], the gamma-glutamyltranspeptidase-activated Ga-IRDye800 [182], and EGFR-targeting Cetuximab-IRDye800CW [184]. The EGFR-targeted ABY-029 recently posted results (NCT02901925 accessed 16 January 2025) but is yet to appear in the literature.

### 6.2. Enhanced Fluorescence

Efforts to improve sensitivity have focused on the development of fluorophores, which emit in the NIR and NIR II spectrum [184,189] to improve tissue penetration and reduce autofluorescence. Improving penetration of the BBB has also been shown to improve sensitivity through increased fluorescence levels [187].

Nanoparticles such as quantum dots are also being explored [193]. These can be engineered to include surface coatings and negative surface charge to prevent serum protein binding, improving both half-life and accumulation via EPR at areas of the BBB breakdown. Despite the lack of specific targeting, this technique has shown early promise in glioblastoma, albeit with some concerns around the sensitivity at tumour margins [194,195]. At present nanotracers remain limited to preclinical studies for FGS but have commenced clinical studies as drug delivery and treatment adjuncts [196].

### 6.3. Barriers to Novel Fluorophores

The challenges in development of novel fluorophores include high research and development costs [197], molecular heterogeneity [198], the paucity of preclinical disease models [199], safety concerns [193] and the relative lack of findings in glioma [200]. FDA approval also typically requires overall survival to be assessed, which is problematic for FGS agents given the heterogeneity of treatment and outcomes based on tumour location, size, molecular characteristics, and fitness for post-operative systemic therapies. The 5-ALA pathway is instructive for the development and clinical translation of novel fluorophores and highlights the significant evidence and time required for approval [14]. To date, the few phase II clinical trials of novel fluorophores have been either lacking or equivocal.

### 6.4. Future Prospects

The development of novel fluorophores for glioma FGS has been dealt a blow by recent early phase trial results (NCT03579602 accessed 16 January 2025). Nevertheless, the need for agents that allow the operating field to be safely visualised whilst simultaneously labelling pathological tissue with high sensitivity and specificity remains pressing. The recent award of the Nobel Prize in chemistry for the discovery and development of quantum dots in 2023 highlights the huge potential for nano-molecules in this field. This is synergistic with developing techniques such as ultrasound [201] for focal blood–brain barrier disruption, which is a formidable obstacle to the delivery of new agents to the surgical field. Beyond challenges of safety and specificity described above, clinicians keen to explore novel agents in their practice, translational researchers, and glioma funders should be conscious of the formidable regulatory issues and time required to bring these agents into routine use.

## 7. Conclusions

Fluorescence guidance has become an established and widely accepted adjunct in glioma surgery, with demonstrable benefits to the extent of resection.

Of the fluorophores in current clinical use, 5-ALA is alone in receiving regulatory approval for HGG surgery in the USA and Europe, backed by RCTs showing increased extent of resection. However, FGS in LGG is limited by low levels of visible fluorescence. All FGS techniques remain constrained by the challenges of detection.

Pharmacokinetic studies highlight the potential for FGS techniques and fluorophores to be optimised for administration timing and dose. The development of pathology-specific, standardized protocols for dose, administration timing, and detection techniques is essential, particularly in the case of fluorescein, where significant variation in practice can be seen across the literature, significantly undermining the evidence bases for its use. With increasing emphasis on molecular markers in glioma diagnosis in the updated 2021 World Health Organisation (WHO) classification of tumours of the central nervous system, the possibility for fluorescence to be studied in molecular rather than histologically defined subgroups is an exciting ongoing area of study.

Current methods of FGS are impacted by subjective interpretation, the inability to safely operate under excitation light, non-linear tissue optical distortions, overlapping fluorescence, and, in particular subgroups, low emission. Exciting work is underway to overcome these limitations, with promising integrated quantitative fluorescence systems for glioma surgery. This work has been spearheaded using point probes in conjunction with 5-ALA and is expanding to include wide-field techniques that can more seamlessly integrate with surgical workflows. Synergistic developments in detection hardware, including hyperspectral imaging cameras with novel AI-driven analyses hold the potential to quantify specific fluorophores and understand the relevance of their changing spectral signatures within the heterogeneous glioma micro-environment. Confocal microscopy gives the potential for in vivo virtual biopsies integrated with wide-field infiltrative margin detection.

All established fluorophores covered in this review are limited either by their inherent molecular properties or mechanism of action. The development of novel, tumour-targeted fluorophores with tailored fluorescence characteristics holds great promise, but clinicians should be aware of the significant timelines involved in development, validation, and regulatory approval and disappointing Phase 1 results.

As fluorescence-guided surgery develops, the development of consensus guidelines for evaluation is crucial. These are beginning to emerge in the technical [202] and clinical [53] literature. Researchers and clinicians should understand the distinction between PPV, NPV, sensitivity, and specificity. The development of measurable, relevant, and accepted outcome measures in clinical trials remains a challenge across the glioma literature, and patient-specific measures such as deficit-free survival are increasingly considered essential [203].

## Figures and Tables

**Figure 1 cancers-17-02019-f001:**
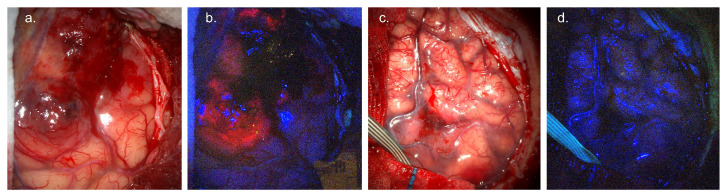
5-ALA fluorescence-guided surgery. (**a**) White-light visualisation of glioblastoma involving cortical surface. (**b**) Surgical field under blue (405 nm) light, demonstrating avid red/pink tumour fluorescence. (**c**) White-light visualisation of WHO Grade 2 oligodendroglioma involving cortical surface. (**d**) Blue-light imaging, demonstrating lack of visible tumour fluorescence. *Images courtesy of Neuro PPEye study team, King’s College London, UK (NCT05397574)*.

**Figure 2 cancers-17-02019-f002:**
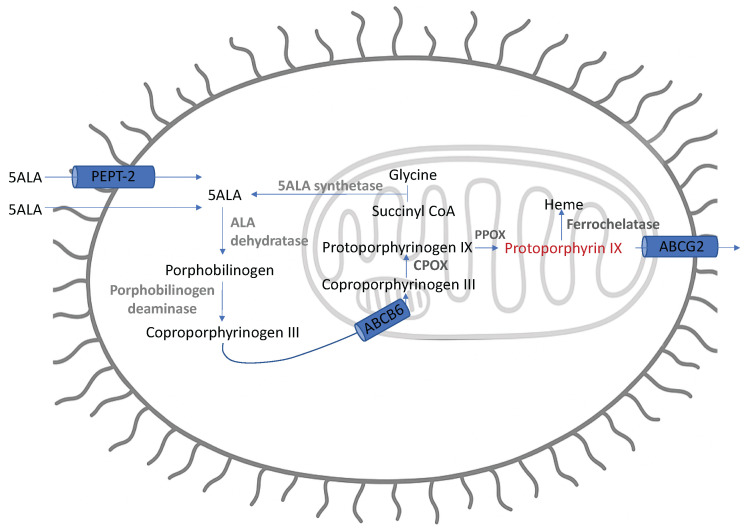
Simplified schematic of PpIX synthesis in heme biosynthetic pathway. ABCG2: ATP-binding cassette G2; ABCB6: ATP-binding cassette B6; CPOX: coproporphyrinogen oxidase; PPOX: protoporphyrinogen oxidase.

**Figure 3 cancers-17-02019-f003:**
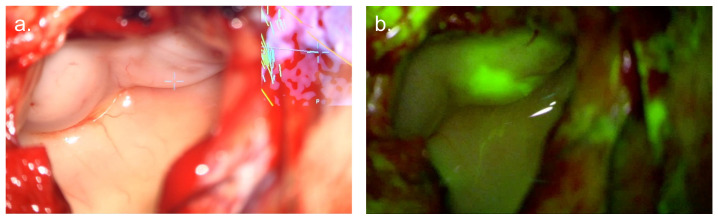
Fluorescein fluorescence guided surgery (**a**) White-light visualisation of infiltrated gyrus. (**b**) Surgical field under yellow (560 nm) light, demonstrating green fluorescence at areas of tumour infiltration. *Images courtesy of Mr Ryan Mathew, University of Leeds, UK*.

**Figure 4 cancers-17-02019-f004:**
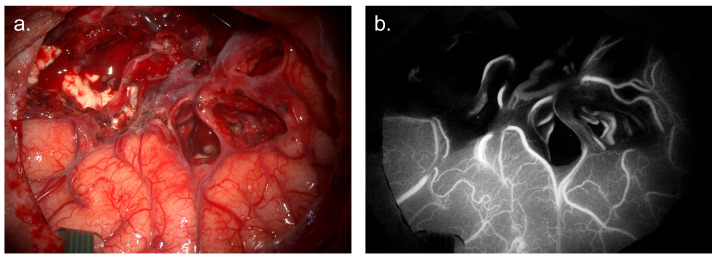
ICG angiography during glioma surgery. (**a**) White-light visualisation of resection cavity showing skeletonized cortical and tumour associated vessels. (**b**) IR imaging of surgical field with 800 nm excitation light, demonstrating abnormal tumour vessels and preserved flow in overlying cortical vessels. *Images courtesy of Neuro PPEye study team, King’s College London, UK (NCT05397574)*.

**Figure 5 cancers-17-02019-f005:**
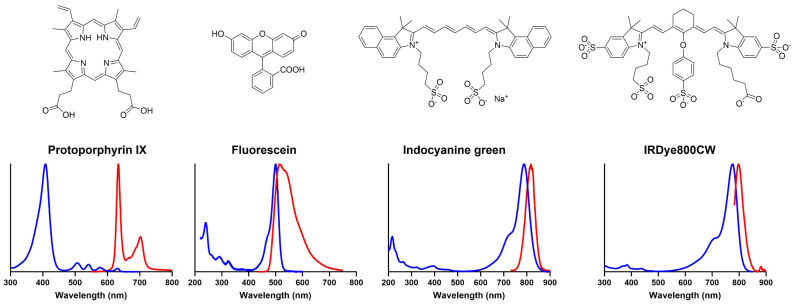
Chemical structures and optical emission spectra of key fluorophores used in, and under investigation for, glioma FGS [188]. Excitation spectra are shown in blue, and emission spectra are shown in red. *IRDye800CW spectra courtesy of Mr Jean-Romain Lotthe, Kings College London*.

## Data Availability

No new data were created or analysed in this study. Data sharing is not applicable to this article.

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
