# Peer review of "Fluorescence Guidance in Glioma Surgery: A Narrative Review of Current Evidence and the Drive Towards Objective Margin Differentiation"

_cancers, 2025, doi:10.3390/cancers17122019_

Round 1

Reviewer 1 Report

Comments and Suggestions for Authors

The review titled “Fluorescence guidance in glioma surgery: A narrative review of current evidence and the drive towards objective margin differentiation” discusses the use of fluorescent guidance in the surgical removal of gliomas. The authors outline the indications for this approach, the required equipment, how it works, the supporting evidence, as well as its limitations and regulatory concerns associated with each fluorophore currently used in clinical settings. Furthermore, the review introduces new systems for visualizing fluorescent signals and explores the practical applications of novel fluorescent dyes. Overall, the writing is clear and well-organized. Despite the perfect quality of the work, there are several areas that need improvement, as described below:

  1. In the introduction, what is new about your review compared to previous ones (10.3389/fneur.2021.682151, 10.1093/neuros/nyaa475, 10.3390/cancers16152698)?
  2. Why is the fluorescein UV-Vis spectrum noisy at low concentrations in Figure 5?
  3. After sections, 5 and 6, small conclusions on the prospects for the development of new fluorescent dyes and fluorescence signal detection systems are missing.

Author Response

 We thank you for your careful reading of our manuscript and your helpful comments. Please see our attached letter containing a detailed response, and outlining the amendments based on your kind feedback.

Based on your suggestions we have made a number of changes to the manuscript as outlined below. These serve to highlight the novelty of the current review, specifically with regards to work over the last 12 months suggesting an unanticipated mechanism of action of 5-ALA and somewhat disappointing trial results in novel fluorescence agents. As we have clarified in our updated introduction we believe this highlights the opportunity to derive new intra-operative information on the glioma micro-environment using novel imaging technologies with fluorescence agents already in routine use (in addition to increasing their sensitivity and specificity). It also serves to highlight the difficulty and signifi cant timescales in developing novel fluorescence agents. We believe these twin developments are of critical importance to both clinicians and researchers involved in the ongoing development of fluorescence guided glioma surgery, and to funders of translational research. As suggested in comment 3, this has also been highlighted in the conclusions to sections 5 and 6 as detailed below in our reviewer response letter.

Reviewer 2 Report

Comments and Suggestions for Authors

This manuscript is a very well written and thorough review covering the current progress on fluorescence guided surgery in glioma. This manuscript nicely focused on the practical application in clinic, included approved methods already in use and new methods under development, which will be valuable information to the neurosurgery field. The authors also nicely summarized the known mechanisms of each approach, which will be valuable information to the biomedical field.

A few minor suggestions:

1) 2.1.2., the authors may consider merge all paragraphs and try not to leave one sentence as a paragraph.

2) In each section, Dose and Administration and timing, such as 2.1.2, and 2.1.3; 3.1.2 and 3.1.3, are mostly the same topic: the timing and dose of administration. The authors may consider merge them.

3) Page 6, line 204; page 10, line 382, please provide the full name for FLAIR, NICE as it’s the first time mentioned.

4) Page 12, line 427. Could the authors add some explanation on ‘second window effect’?

5) Page 17, line 609. “WHO 2021 WHO grading”.

Author Response

We thank you for your careful reading of our manuscript and your helpful comments. Based on your suggestions we have made modifications to the review structure, including consolidation of the Dose, Administration and Timing Sections. We are also grateful for your highlighting of terms that in the original manuscript were either missing from the glossary, or missing comprehensive explanations - which we have addressed as detailed below in our attached letter.

Reviewer 3 Report

Comments and Suggestions for Authors

The review paper submitted by Elliot et al. discuss about the latest research in the field of fluorescence guidance in glioma surgery. The authors presented the most important fluorescence agents by describing not only their advantages but also their limitations. Moreover, the clinical trials are also discussed. 

In my opinion this review is very interesting and helpful for both the practitioners and researchers. I have only one comment:

1. the authors must add some perspectives and also some critical personal remarks.

Author Response

We thank you for your careful reading of our manuscript and your helpful comments, which have highlighted key areas for the manuscript to be improved. In particular based on your comment we have expanded the introduction and discussion, in addition to adding short conclusions to sections 5 and 6. These highlight the important of recent development suggesting an unexpected mechanism
of action in 5ALA which we believe highlights the opportunity to derive new intra-operative information on the real time glioma micro-environment using novel imaging technologies with fluorescence agents already in routine surgical use. We believe this, coupled with disappointing early results for novel fluorescence agents in humans as outlined in section 6 is of importance both to clinicians integrating fluorescence guided surgery to their practice, and researchers and funders involved in the ongoing development of fluorescence guided glioma surgery. Detailed changes are outlined in our attached letter.
